

# Quality assessment of the Ozone_cci Climate Research Data Package (release 2017): 1. Ground-based validation of total ozone column data products

Katerina Garane[1], Christophe Lerot[2,] Melanie Coldewey-Egbers[3], Tijl Verhoelst[2], Irene Zyrichidou[1], Dimitris S. Balis[1], Thomas Danckaert[2], Florence Goutail[4], Jose Granville[2], Daan Hubert[2], Maria Elissavet Koukouli[1], Arno Keppens[2], Jean-Christopher Lambert[2], Diego Loyola[3], Jean-Pierre Pommereau[4], Michel Van Roozendael[2] and Claus Zehner[5]

[1] Laboratory of Atmospheric Physics, Aristotle University of Thessaloniki, Thessaloniki 54124, Greece.
[2] Royal Belgian Institute for Space Aeronomy (BIRA-IASB), 3, Avenue Circulaire, B-1180 Brussels, Belgium.
[3] Deutsches Zentrum für Luft- und Raumfahrt (DLR), Institut für Methodik der Fernerkundung (IMF), 82234 Oberpfaffenhofen, Germany.
[4] LATMOS, CNRS, University Versailles St Quentin, Guyancourt, France
[5] European Space Agency, ESRIN, Frascati, Italy

*Correspondence to*: Katerina Garane (agarane@auth.gr)

**Abstract.** The GOME-type Total Ozone Essential Climate Variable (GTO-ECV) is a Level-3 data record, which combines individual sensor products into one single cohesive record covering the 22 year period from 1995 to 2017, generated in the frame of the European Space Agency's Climate Change Initiative Phase-II. It is based on Level-2 total ozone data produced by the GODFIT v4 algorithm as applied to the GOME/ERS-2, OMI/Aura, SCIAMACHY/Envisat and GOME-2/MetopA and /MetopB observations. In this paper we examine whether GTO-ECV meets the specific requirements set by the international climate-chemistry modelling community for decadal stability, long-term and short term accuracy. In the following, we present the validation of the 2017 release of the Climate Research Data Package Total Ozone Column (CRDP TOC), both at Level-2 and Level-3. The individual Level-2 data sets show excellent inter-sensor consistency with mean differences generally within 0.5 % at moderate latitudes (+/- 50°), whereas the Level-3 data sets show mean differences with respect to the OMI reference data record that span between -0.2 ± 0.9 % (for GOME-2B) and 1.0 ± 1.4 % (for SCIAMACHY). Very similar findings are reported for the Level-2 validation against independent ground-based TOC observations reported by Brewer, Dobson and SAOZ instruments; the mean bias between GODFIT v4 satellite TOC and ground instrument is well within 1.0 ± 1.0 % for all sensors, the drift per decade spans between -0.5 % to 1.0 ± 1.0 % depending on the sensor, and the peak-to-peak seasonality of the differences ranges between ~1% for GOME and OMI, to ~2% for SCIAMACHY. For the Level-3 validation, as a first step the aim was to show that the Level-3 CRDP produces consistent findings as the Level-2 individual sensor comparisons. We show an excellent agreement with 0.5 to 2 % peak-to-peak amplitude for the monthly mean difference time series and a negligible drift per decade in the Northern Hemisphere differences at -0.11 ± 0.10 % per decade for Dobson and +0.22 ± 0.08 % per decade for Brewer collocations. The exceptional quality of the Level-3 GTO-ECV v3 TOC record temporal stability well satisfies the requirements for the total



ozone measurement decadal stability of between 1 – 3 % and the short term and long-term accuracy requirements of 2% and 3% respectively, showing an excellent inter-sensor consistency both in the Level-2 GODFIT v4 as well as in the Level-3 GTO-ECV v3 datasets and thus can be used for longer term analysis of the ozone layer, such as decadal trend studies, chemistry-climate model evaluation and data assimilation applications.

**1    Introduction**

The European Space Agency's *Climate Change Initiative* Phases-I & -II focused on building consolidated climate-relevant Ozone data sets as Essential Climate Variables, ECVs. During Phase-I, the Ozone CCI mostly concentrated on developing and demonstrating improved algorithms and methods, with the aim to define new baselines for the generation of consistent, state-of-the-art and fully characterized long-term ozone data products derived from a complete suite of European nadir and

limb-type sensors. For the first time, Earth Observation science teams consisting of leading experts from European ozone sensing communities were gathered in a single project working towards common objectives defined against requirements formulated by the scientific user community. This resulted in new synergies, exchanges of ideas, and overall significant progress in terms of data harmonisation and understanding of quality issues at Level-1, Level-2 and Level-3. Three lines of multi-sensor ozone data products were hence developed: i. total ozone columns from UV nadir instruments, ii. low resolution

ozone profiles from nadir sensors and iii. stratospheric and upper tropospheric ozone profiles from limb and occultation types of sensors. During Phase-II, existing state-of-the-art ozone retrieval algorithms were further developed and applied to long time series of observations from all relevant ESA atmospheric chemistry sensors, with the aim to generate well characterized and validated ozone data products that meet as closely as possible the requirements formulated by the Global Climate Observing System (GCOS) as well as the Climate Modelling User Group (CMUG) climate modelling community,

for ozone column and profile ECVs. The most important user requirements were identified as: i. homogenized multi-decadal records, ii. records with good vertical resolution in the (lower) stratosphere and iii. records with good horizontal resolution in the troposphere, the main gap being the lack of multi-decadal high-vertical resolution ozone profile data sets that cover the full ozone depletion time period (1980-present) and provide a potential to cover the upcoming ozone recovery time period.

This work addresses the first of these requirements, the Level-2 and Level-3 homogenized multi-decadal total ozone CRDP,

with two more companion papers (Keppens et al., 2017; Hubert et al., 2017) expanding on the limb and nadir ozone profile CRDPs. On total ozone, 21 years of harmonised Level-2 data records from GOME/ERS-2, OMI/Aura, SCIAMACHY/Envisat and GOME-2/MetopA and /MetopB sensors have been produced using an advanced version of the direct-fitting GODFIT v4 algorithm. The ESA-CCI total ozone CRDP includes the Level-2 products for each instrument (over the entire instrument lifetime) and a Level-3 merged monthly mean gridded data set using GOME and OMI as long-

term stability reference.

In the following section, we briefly present the GODFIT v4 algorithm that creates the Level-2 CRDPs, followed by the validation against the Brewer, Dobson and SAOZ ground-based instruments and the comparison to the independent Solar



Backscatter Ultraviolet measurements (SBUV) v8.6 long term TOC record. Thereafter, the algorithm that merges the individual Level-2 TOC records to create the Level-3 dataset is presented, followed by the validation to the ground-based records and inter-comparison to the individual Level-2 validation findings. Summary and conclusions are given in the last section.

## 2 Level-2 Total Ozone Columns

### 2.1 Satellite Total Ozone Column records

GODFIT (GOME-type Direct FITting) is an algorithm jointly developed by BIRA-IASB, RT Solutions and DLR to retrieve Total Ozone Columns (TOC) from satellite-borne nadir-viewing hyperspectral spectrometers, such as GOME(-2), SCIAMACHY and OMI. It relies on a non-linear least-squares minimization procedure, during which sun-normalized radiances simulated in the Huggins bands (325-335 nm) with the Radiative Transfer model LIDORT (Spurr et al., 2013) are adjusted to the Level-1 measurements. As part of the phase-I of the ESA Ozone_cci project, version 3 of GODFIT has been successfully transferred to other nadir sensors and is comprehensively described in Lerot et al. (2014) and validated in Koukouli et al. (2015). During the second phase of this project, a number of algorithmic improvements have been realized and the full time series of GOME, OMI, SCIAMACHY and GOME-2A/B have been entirely reprocessed with the latest version (v4) of GODFIT. The most important update is the adaptation of the L1 soft-calibration scheme in order to restore the full independency of the satellite observations with respect to the ground-based measurements. This algorithm, described in detail in Danckaert et al. (2017), is also the future baseline for generating the offline operational total ozone from the TROPOMI/S5-p instrument, that launched in October 2017.

The radiance simulations require that the atmosphere is properly defined at each iteration within the retrieval and so a series of auxiliary data are also required. Ozone vertical profiles are prescribed by the total ozone classified climatology recently released by Labow et al. (2015) using MLS and sondes data, combined with the tropospheric column database constructed by Ziemke et al. (2011). The ozone absorption is modelled using the temperature-dependent cross-sections measured by Serdyuchenko et al. (2014). The temperature in each atmospheric layer is prescribed by a priori profiles, allowed to be shifted by a constant offset, determined simultaneously to the total column. All cross-sections are preconvolved at the respective instrumental resolution and an improved correction for the so called solar $I_0$-effect (Aliwell et al., 2002) has been applied (Danckaert et al., 2017). GODFIT has the capability to characterize instrumental slit function on an orbit-basis by fitting pre-determined functions such as (Super-)Gaussian shapes (Beirle et al., 2017) or by stretching slit functions pre-measured on-ground. To account for contamination by clouds and/or aerosols, an effective scene approach is used (Coldewey-Egbers et al., 2005) in which the effective albedo of a scene located in between the cloud top height and the ground surface is fitted during the retrieval. The altitude of this effective scene depends on both the effective cloud fraction and cloud top altitude provided by independent cloud algorithms (FRESCO v7, Wang et al., 2008 or the O2-O2 product, Veefkind et al., 2016). Radiances are simulated on-the-fly with the scalar radiative transfer model LIDORT for GOME,





SCIAMACHY and GOME-2. Because of the heavy computational burden of those simulations, the radiances may alternatively be extracted from a pre-computed look-up table, of which the granularity has been cautiously defined in order to limit interpolation errors while keeping a reasonable size (Danckaert et al., 2017). Once simulated, correction terms are applied to the radiances to correct for the impact of atmospheric polarization and inelastic scattering processes (Lerot et al., 2014).

When a common retrieval algorithm is applied to various instruments, systematic differences may remain due to calibration deficiencies or instrumental degradation effects affecting the Level-1 reflectance data. To generate the CCI total ozone data sets with the high inter-sensor consistency required for climate studies, an original soft-calibration scheme had been incorporated within GODFIT v3. This procedure, extensively described in Lerot et al. (2014), relied on reference total column measurements at selected Northern mid-latitude Brewer stations. Although it was shown to work well, this approach had the disadvantage to introduce a link between the satellite and ground-based measurements. As illustrated in Figure 1, experience has shown that the GOME and OMI sensors perform in an extremely stable way and do not require any spectral soft-calibration procedure. Therefore it was decided to use these two instruments to soft-calibrate the spectra measured by SCIAMACHY and GOME-2A/B. In practice, for every cloud-free satellite pixel falling into a reference sector between 40° S-50° N and 175° W-145° W, the closest reference clear-sky OMI (or GOME before 2005) column is used to simulate a radiance (using the GODFIT forward model), which is then compared to the Level-1 spectrum recorded by the sensor to be soft-calibrated. Such comparisons are done systematically for a large number of pixels (e.g. several hundreds of thousands for GOME-2A) spanning most of the observation geometries and the full time series, which allows to identify and correct for systematic issues in the Level-1 data. See Lerot et al. (2014) for more details on the soft-calibration approach.

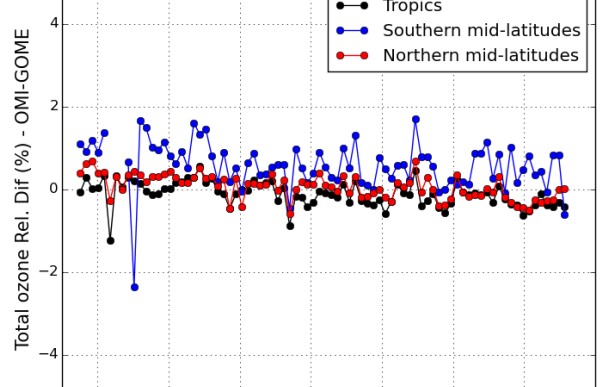

**Figure 1. Time series of the relative differences between the total ozone columns retrieved from the GOME and OMI sensors for different latitude bands. Retrievals have been performed without any soft-calibration of the reflectances for both instruments.**





Using this new GODFIT v4 baseline, the time series of GOME, SCIAMACHY, GOME-2A/B and OMI have been entirely reprocessed. Figure 2 illustrates the excellent consistency between the individual Level-2 data sets with mean differences generally within 0.5% at moderate latitudes (+/-50°). The Level-2 data sets are publicly available on the Ozone_cci website (http://www.esa-ozone-cci.org) and the time series are also regularly extended as part of the Copernicus Climate Change

Service (C3S).

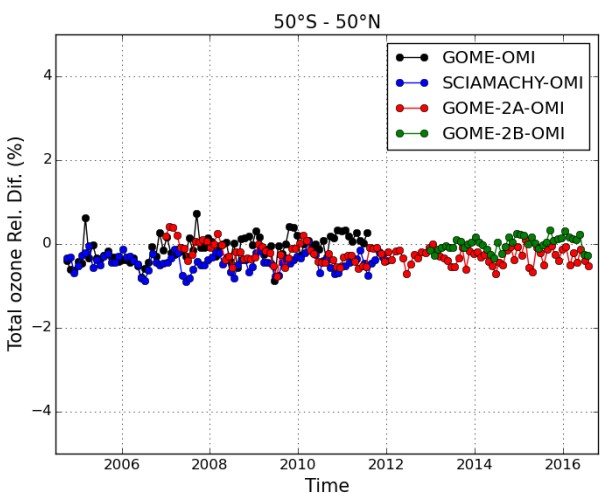

**Figure 2. Time series of the relative differences between the total ozone columns retrieved from GOME, SCIAMACHY and GOME-2A/B with respect to OMI.**

**2.2      Ground-based Total Ozone Column records**

For the purposes of this work, both direct-sun measurements (from Dobson and Brewer UV spectrophotometers) and zenith-sky scattered-light (ZSL-DOAS) measurements were used as ground-based reference data.

Total ozone column measurements from Dobson and Brewer UV spectrophotometers, were downloaded from the WOUDC archive (http://www.woudc.org), see Table S2 & S3 for a complete list. The measurement techniques and the data analysis

methodology are extensively analyzed in Koukouli et al. (2015) and in references therein. It is important to point out that according to Van Roozendael et al. (1998), the estimated total uncertainty for the Dobson spectrophotometer is about 1 % for cloud-free direct Sun observations and 2 – 3 % for zenith-sky or cloudy observations, while the error of individual total ozone measurements for a well-maintained Brewer instrument is about 1 % (e.g. Kerr et al., 1988).

The main issues that have to be taken into account during the validation process with these direct-sun instruments are: (a)

TOC measurements from Dobsons spectrometers depend on the stratospheric effective temperature, which is manifested in the comparisons as a seasonality effect (Basher, 1982; Bernhard et al., 2005), (b) even though the principles of operation



between Dobsons and Brewers do not differ significantly, TOC measurements from the two types of instruments show small differences in the range of ± 0.6 % due to the use of different wavelengths and the different temperature dependence for the ozone absorption coefficients (Staehelin et al., 2003) and (c) due to the limited number and poor spatial distribution of stations with Brewer instruments in the Southern Hemisphere (all of them allocated in the Antarctic), the Dobson network is

considered much more suitable to investigate spatial homogeneity of satellite products below the Equator.

TOC ground-based measurements from the abovementioned instruments have been extensively used in past publications for the purpose of analysis and validation of satellite data (see for e.g. Balis et al, 2007a; Balis et al, 2007b; Antón et al., 2009; Loyola et al., 2011; Koukouli et al., 2012; Labow et al., 2013; Bak et al. 2015; Koukouli et al., 2015). The ground-based stations were selected in accordance with the criteria discussed in detail in Balis et al. (2007a) and Balis et al. (2007b). Their

measurements are thoroughly inspected once a year, in the aspect of quality assurance and stability, following the principles described in Fioletov et al. (1999); Vanicek (2006) and Fioletov et al. (2008), among others.

The GODFIT v4 total ozone columns were also compared against twilight zenith-sky measurements obtained with ZSL-DOAS (Zenith Scattered Light Differential Optical Absorption Spectroscopy) instruments. Most of these instruments form part of the SAOZ network (Système d'Analyse par Observation Zénitale; Pommereau & Goutail, 1988) of the Network for

the Detection of Atmospheric Composition Change (NDACC). In NDACC, four slightly different ZSL-DOAS instruments are also routinely reporting data (see Table S1 for complete list of instruments used). The total accuracy of ZSL-DOAS measurements, including the cross-section uncertainties, is considered to be of the order of 6 % (Hendrick et al., 2011). To avoid confusion in the paper, hereafter they will all be referred to as "SAOZ measurements".

These twilight zenith-sky measurements are complementary to the Brewer and Dobson measurements for several reasons:

(a) they use spectral features of the visible Chappuis band, where the ozone differential absorption cross sections are temperature insensitive, (b) the long horizontal stratospheric optical path allows measurements of the column above cloudy scenes, and (c) measurements are always performed in the same, small, SZA range (86° - 91°). For further details on the measurement procedures and on the specific collocation approach, taking into account the actual area of measurement sensitivity, we refer to Balis et al. (2007a), Koukouli et al. (2015), and references therein. After quality control and the

application of thresholds on the minimum number of collocated measurements, data from about 20 instruments were used, covering both the Northern and Southern hemisphere up to high latitudes and leaving only the equatorial region poorly sampled (see Figure S1 for the locations of all three types of instruments). In spite of the dedicated collocation method, some residual errors due to co-location mismatch may persist and must be kept in mind, in particular at high latitudes, as shown by Verhoelst et al. (2015).

## 30   2.3    Level-2 validation results and discussion

As a basis for the validation process of the satellite TOC measurements, pairs of co-located satellite and ground-based measurements are formed and their percentage difference is calculated. Specific criteria are applied to minimize the noise of the comparison:





i.  For the Dobsons and Brewers: (a) the maximum search radius between the ground-based stations and the center coordinates of the satellite pixel is set to 150 km and the spatially closest satellite observations are paired with the ground-based station's daily mean measurement and (b) only direct-sun ground-based measurements are used for the validation process, since they are deemed to be most accurate.

ii. For the SAOZ measurements, the large displacement (with respect to the instrument location) of the actual measurement sensitivity is taken into account by requiring satellite pixels to intersect with a 2-D (lat, lon) polygon describing the true area of measurement sensitivity, see Balis et al. (2007a) and Verhoelst et al. (2015) for full details.

Following those criteria, three timeseries (one for each type of ground-based instrument) of the percentage difference, are formed. Hereupon, a statistical analysis of the timeseries is performed, separately for each type of instrument, so as to study a variety of possible dependences on geospatial parameters such as the season, latitude, observation geometry, etc. The results of the analysis are shown in the following graphs and are summed up in Table 1. In the figures presented in this section, the dependency of the percentage difference between satellite and ground-based TOC measurements on parameters, such as the ones mentioned above, is displayed (the line colors used for Figure 3 to Figure 6 are:  GOME → black line; SCIAMACHY → blue line; OMI → cyan line; GOME-2A → green line and GOME-2B→ orange line). It should be noted that Southern Hemisphere GOME measurements are only shown before 2003, when it encountered downlink telemetry problems.

In Figure 3 the timeseries of the percentage difference between the monthly mean TOC measurements from five different satellites to the co-located Dobson, Brewer and SAOZ ground-based measurements are shown. In all panels the entire available timeseries from each satellite instrument is displayed (except GOME for the Southern Hemisphere, as mentioned above). The comparison with the Dobson measurements is presented in panel (a), which corresponds to the Northern Hemisphere (NH) stations, and panel (b), which presents the Southern Hemisphere (SH) percentage differences. It is shown that the NH timeseries are highly consistent and stable for all five satellites, with an amplitude of ~ 2 % for all sensors apart from SCIAMACHY, which shows a slightly increased variability with certain months under-estimating the ground-based mean (differences reaching -1 %). Part of the seasonality observed in Figure 3 – panels (a) and (b), is due to the known Dobson dependency on the effective temperature of the stratosphere (Koukouli et al., 2016). The ~ 1.5 % bias of the satellite TOCs compared to the Dobson TOCs might be related to systematic uncertainties in the different ozone absorption cross-sections used to retrieve satellite and ground-based measurements which have been estimated to rise up to ±2% (Orphal et al., 2016).

The comparison for the SH Dobson measurements (Figure 3, panel b) is showing higher variability due to the fact that the number of available stations in this part of the globe is limited and their measurements are greatly affected by the vigorous phenomena developing over the Antarctic. However, all timeseries present a rather consistent and stable behavior, similar to that shown in the NH, with a bias of the order of 1 - 1.5 % for OMI, GOME-2A and GOME-2B.



(a)

(d)

(b)

(e)

(c)

**Figure 3. The time series of the monthly mean percentage differences between the five satellite instruments and the co-located ground-based TOC measurements performed by Dobsons (panel a: Northern Hemisphere and b: Southern Hemisphere), Brewers (panel c: Northern Hemisphere) and SAOZ (panel d: Northern Hemisphere and e: Southern Hemisphere) instruments.**



In Figure 3 - panel (c), the same plot of the percentage differences between the satellites and Brewer ground-based measurements performed at stations located in the NH, is shown. Due to the extremely limited number of stations with Brewer spectrophotometers in the SH, positioned exclusively on the Antarctic, it was decided not to present the respective plot. It is evident that the consistency and the stability of the satellite measurements is excellent for the whole time period of

available data and for the whole set of five sensors. In this case, the overall bias of the comparison is up to 1 % for GOME, 0 % for SCIAMACHY and 1.5 % for the rest of the instruments, with peak-to-peak amplitude of the order of 1 – 2.5 %.

Panels (d) and (e) of Figure 3, depict the timeseries of the comparison to the SAOZ network, for the Northern and Southern Hemisphere, respectively. Even though the seasonality effect, which is known to be present in comparisons between SAOZ and direct-sun measurements (Hendrick et al., 2011) , is obviously stronger in these figures than in the other three panels, the

inter-sensor consistency is evident here as well. The overall bias of the SAOZ comparison is fairly stable at 1.5% in the NH, but rather variable for the SH, which can be attributed to the large number of high-latitude stations contributing to the SH statistics. SAOZ comparisons at these stations are known to be affected more by co-location mismatch (Verhoelst et al., 2015).

Following, the dependence of the percentage differences of the five satellites measurements to the ground-based TOC

measurements, on solar zenith angle (SZA) was investigated, as shown in Figure 4. We have chosen to present this plot only for Brewer (panel a – NH) and SAOZ (panels b – NH and c – SH) measurements, due to the aforementioned dependency of the Dobson measurements to the stratospheric effective temperature, which is highly correlated with SZA. Firstly, as it is seen in Figure 4, all curves in each plot have highly consistent dependencies on SZA, which proves that, irrespective of its magnitude, the dependence can be contributed mainly on the ground based measurements of each kind.

Specifically, panel (a) shows that the percentage difference of the measurements is almost constant for the Brewer comparison and it is only increasing for SZAs larger than 70°. SCIAMACHY however shows a slightly stronger dependence on SZA starting from low angles. Comparisons performed at SZAs over 75° and below 25° are affected by the limited number of observations and the uncertainties of the ground-based measurements themselves. Hence it is difficult to assess their significance level. In Figure 4 – panels (b) and (c), we show that the SZA dependence between satellite and SAOZ

ground measurements was up to 4 % at the highest satellite-viewed SZAs (>80°) at all high-latitude stations, irrespective of season. There was also some minor dependence at very small SZAs in the Northern Tropics, but this is based on only a few tropical stations with limited data, and it is not confirmed by the Brewer comparisons. There are also some systematic inter-hemispheric differences for SAOZ measurements, which is obvious when comparing panels (b) and (c) of Figure 4, in particular due to comparisons at some Northern high-latitude stations being biased high (up to 5%), and those at Southern

high-latitude stations being biased low (of the order of 2%), as shown in Figure 5 – panel (c) that will be commented on below.





(a)

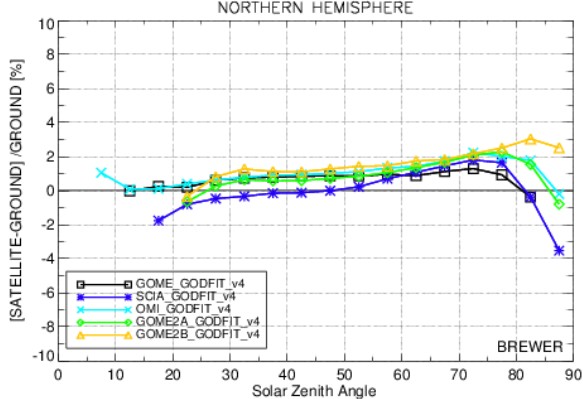

(b)

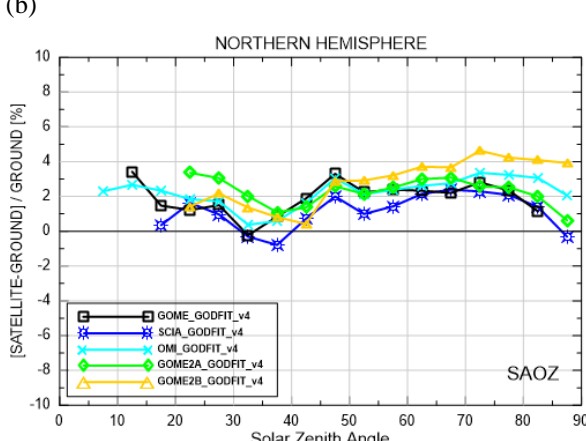

(c)

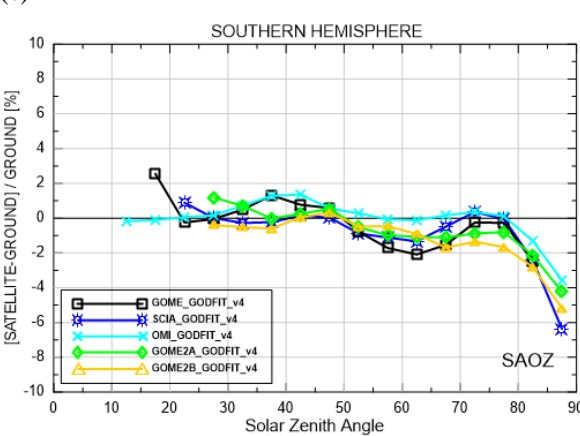

**Figure 4. The dependence of the percentage differences between the satellites TOC measurements to the measurements of the Brewer (panel a, NH only) and SAOZ (panels b-NH and c-SH) ground-based stations, on solar zenith angle.**



(a)

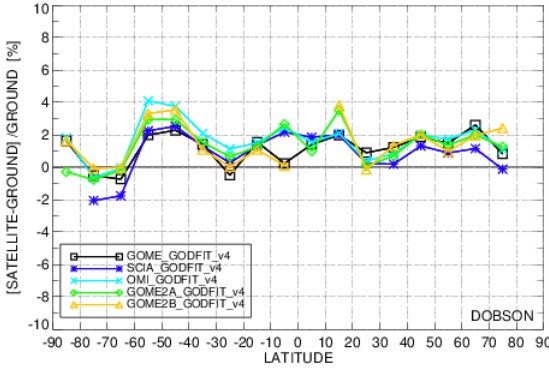

(b)

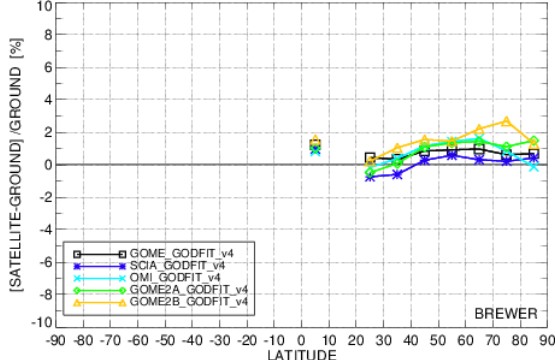

(c)

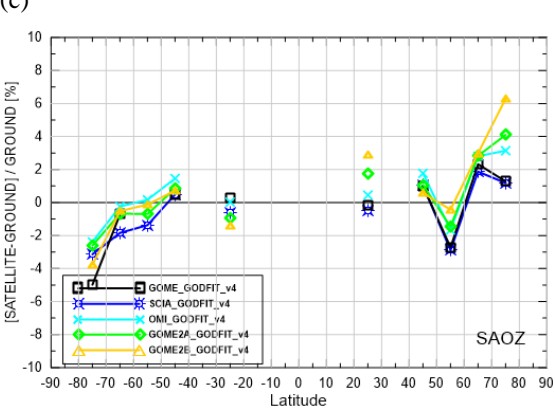

**Figure 5: The percentage difference between the five satellites TOC measurements and ground-based measurements from Dobson (panel a), Brewer (panel b) and SAOZ (panel c) instruments, as a function of latitude.**



Additionally, the dependency of the satellite and ground-based measurements percentage difference on latitude, is presented. In Figure 5 - panel (a), the ground-based measurements are performed by Dobson spectrophotometers, in panel (b) Brewer data are used, while in panel (c) the comparison with the SAOZ data record is shown. It is obvious in this exercise too, that

all five satellite sensors appear to be very consistent, regardless of the ground-based instrument type, which is the main concern of this work. It is also noticeable that, mainly for Brewer and Dobson ground-based measurements, the dependency on latitude is less eminent for the NH due to the much higher number of collocations found there. Specifically, the comparisons with Dobson measurements show differences between 0 and 2 % for latitudes between -40° and 0° as well as for the entire NH, similar to the Brewer comparisons. In the SH, especially Southwards of -40°, the comparisons show

differences ranging between -2 and 4 %, depending on the satellite sensor, partially attributed to the small number of stations located in that part of the Earth and partially to the higher variability of the TOCs within the Southern polar vortex (see also Verhoelst et al., 2015). In Figure 5 - panel (c), where the comparison with the SAOZ measurements is shown, a higher dependency on latitude is eminent even for the NH, where the other two ground-based instruments have completely different performances. Nevertheless, the inter-sensor consistency is very satisfactory in this comparison too, except for the high

altitude Izana station located at 28°N (near the NH tropics), for which the differences were adjusted to take into account the missing column in the ground-based measurement but some residual effect due to different satellite pixel sizes is probably still present. Also the measurements performed by the stations located in the belt 70° - 80°N show larger differences between sensors, but these discrepancies are not confirmed by the Brewer or the Dobson networks and they are most probably related to the larger (and pixel-size dependent) smoothing difference errors between SAOZ and the satellite measurements.

According to the guidelines given at the Ozone_cci project's User Requirement Document (Version: 2.1) (van der A, 2011), Table 5, the stability of the total ozone column measurements must be among 1 and 3 %/decade, the evolution of the ozone layer (radiative forcing) has to be less than 2 % and the seasonal cycle and inter-annual (short-term) variability should be less than 3 %. To investigate whether the five satellite data records are compliant to those requirements, a statistical analysis of the percentage deviation between satellite and ground-based measurements was performed with the statistics presented in

Table 1. The first column enumerates the physical quantity studied, the second column differentiates between Brewer, Dobson and SAOZ collocations, the third column shows the results of the statistical analysis for GOME/ERS-2, the fourth column for SCIAMACHY/Envisat, the fifth for OMI/Aura, the sixth for GOME-2/MetopA and the seventh for GOME-2/MetopB sensor. The rows of Table 1 depict: (a) the Monthly Mean Bias and standard deviation (1 sigma), (b) the Monthly mean variability, i.e. the variability of the monthly mean standard deviation values calculated by the Root Mean Square

(RMS), (c) the Drift per decade: the decadal drift and associated standard deviation, (d) the Seasonality: the peak-to-peak amplitude of the seasonal variability, (e) the Latitude: the mean bias and standard deviation as calculated by the latitudinal variability plots on a global scale, (z) the Solar Zenith Angle: the mean bias and standard deviation as calculated from the solar zenith angle ranges shown, on a global scale. The values of the Table are all measured in percent and all the quantities



for the Brewer measurements, as well as quantities (a), (b), (c) and (d) for the Dobson measurements are calculated for the NH only.

The percentages listed in Table 1 prove that the products of the GODFIT v4 algorithm for all five sensors fulfill the requirements set by the European Space Agency's Ozone_cci project, since the amplitude of the short term variability
5  (seasonality) is less than 2 % and the maximum drift per decade is equal to -1.37 ± 1.60 %/decade for GOME-2/MetopB, whose time series is only 3.5 years long and as a result its drift/decade cannot be considered statistically significant. For the

**Table 1: Statistics of the comparison between satellite ground-based TOC measurements.**

| | | GOME/ ERS-2 (%) | SCIAMACHY/ Envisat (%) | OMI/ Aura (%) | GOME-2/ MetopA (%) | GOME-2/ MetopB (%) |
|---|---|---|---|---|---|---|
| Monthly mean bias and 1-sigma | Dobson* | 1.62 ± 0.87 | 0.88 ± 1.01 | 1.26 ± 0.81 | 1.20 ± 1.04 | 1.45 ± 1.08 |
| | Brewer* | 0.83 ± 0.51 | 0.43 ± 0.80 | 1.18 ± 0.50 | 1.08 ± 0.75 | 1.59 ± 0.69 |
| | SAOZ | 1.07 ± 1.46 | 0.41 ± 1.00 | 1.00 ± 0.86 | 0.56 ± 1.10 | 0.57 ± 1.02 |
| Monthly mean variability | Dobson* | ±3.16 | ± 3.22 | ± 3.16 | ± 3.30 | ± 3.16 |
| | Brewer* | ± 3.06 | ± 2.92 | ± 2.82 | ± 2.92 | ± 3.08 |
| | SAOZ | ± 2.40 | ± 2.43 | ± 2.25 | ± 2.31 | ± 2.19 |
| Drift per decade | Dobson* | 0.08 ± 0.13 | -0.61 ± 0.33 | -0.41 ± 0.19 | -0.71 ± 0.35 | -1.37 ± 1.60 |
| | Brewer* | 0.21 ± 0.08 | 0.33 ± 0.26 | 0.01 ± 0.12 | -0.61 ± 0.25 | 0.99 ± 1.02 |
| | SAOZ | 0.51 ± 1.92 | -0.14 ± 2.43 | 0.48 ± 1.54 | -1.32 ± 1.82 | -1.00 ± 4.43 |
| Seasonality (peak – to – peak) | Dobson | N/A | N/A | N/A | N/A | N/A |
| | Brewer* | 0.85 | 2.00 | 0.97 | 1.56 | 1.22 |
| | SAOZ | N/A | N/A | N/A | N/A | N/A |
| Latitude | Dobson | 1.15 ± 0.99 | 0.85 ± 1.34 | 1.65 ± 1.20 | 1.33 ± 1.24 | 1.44 ± 1.33 |
| | Brewer* | 0.75 ± 0.29 | 0.17 ± 0.57 | 0.74 ± 0.67 | 0.88 ± 0.71 | 1.50 ± 0.74 |
| | SAOZ | 0.69 ± 2.67 | 1.34 ± 3.14 | 0.22 ± 2.94 | 1.61 ± 4.55 | 0.82 ± 3.18 |
| Solar Zenith Angle | Dobson (<70°) | 1.19 ± 0.48 | 0.79 ± 0.65 | 1.35 ± 0.67 | 1.02 ± 0.73 | 0.97 ± 1.06 |
| | Brewer* (<70°) | 0.67 ± 0.35 | -0.02 ± 0.89 | 0.88 ± 0.49 | 0.70 ± 0.63 | 1.17 ± 0.61 |
| | SAOZ (<70°) | 0.84 ± 0.68 | 0.40 ± 0.57 | 1.17 ± 0.32 | 1.10 ± 0.49 | 0.91 ± 0.55 |
| | Dobson (>70°) | 1.03 ± 1.14 | -0.92 ± 3.29 | 1.37 ± 1.71 | 0.88 ± 1.92 | 1.77 ± 1.45 |
| | Brewer* (>70°) | 0.61 ± 0.88 | -0.12 ± 2.48 | 1.45 ± 1.12 | 1.27 ± 1.42 | 2.55 ± 0.36 |
| | SAOZ (>70°) | 0.49 ± 0.87 | -0.11 ± 1.94 | 1.02 ± 1.06 | 0.16 ± 1.18 | 0.78 ± 0.87 |

10  * NH only





rest of the sensors the maximum drift per decade is less than ±1 %. In conclusion, the statistics presented in Table 1 indicate that the data sets produced by the Ozone_cci GODFIT v4 algorithm for all five sensors under validation are reliable, homogeneous and consistent.

In order to further demonstrate the long-term inter-sensor consistency of the GODFIT v4 Level-2 total ozone columns,
comparisons to the Solar Backscatter Ultraviolet measurements (SBUV) data products are shown. Daily Level-2 overpass files of total ozone column measurements produced by the SBUV v8.6 algorithm for the locations of the ground-based stations, were downloaded from https://acd-ext.gsfc.nasa.gov/Data_services/merged/ and are described by McPeters et al. (2013) and Frith et al. (2014). The instruments and the respective time periods of measurements used for this comparison are: NOAA 14 SBUV/2 (February 1995 to March 2006), NOAA 16 SBUV/2 (October 2000 to May 2014), NOAA 17
SBUV/2 (July 2002 to March 2013), NOAA 18 SBUV/2 (July 2005 to November 2012) and NOAA 19 SBUV/2 (April 2009 to February 2017). As reported by Labow et al. (2013), their measurements were also validated against Brewer and Dobson ground-based measurements, showing an agreement of the order of ± 1 %.

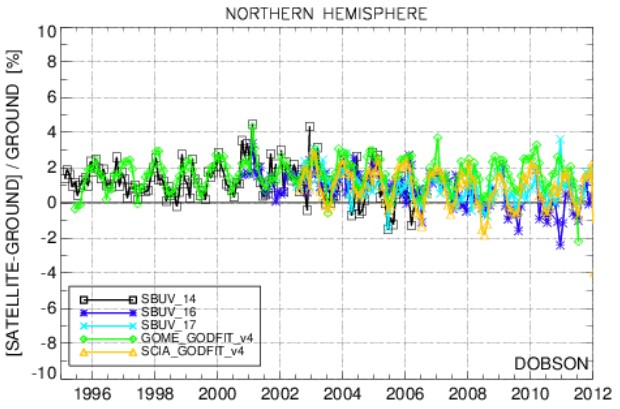
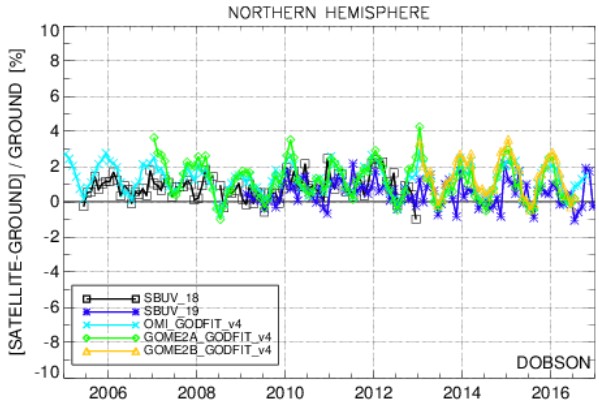

**Figure 6: The timeseries of the percentage differences between satellite and ground-based monthly mean TOC measurements at**
**Northern Hemisphere, separated into two time periods: 1995 – 2012 (left panel) and 2005 – 2017 (right panel). To the left: NOAA**
**14 SBUV/2 (black line), NOAA 16 SBUV/2 (blue line), NOAA 17 SBUV/2 (cyan line), GOME GODFIT v4 (green line) and**
**SCIAMACHY GODFIT v4 (orange line). To the right: NOAA 18 SBUV/2 (black line), NOAA 19 SBUV/2 (blue line), OMI**
**GODFIT v4 (cyan line), GOME-2A GODFIT v4 (green line) and GOME-2B GODFIT v4 (orange line).**

In Figure 6 the percentage deviation of Northern Hemisphere SBUV and GODFIT v4 satellite data sets from the respective ground-based measurements performed by Dobsons, is displayed. In the left panel, the time period 1995 to 2012 is shown, encompassing the available data sets from NOAA 14 SBUV/2, NOAA 16 SBUV/2, NOAA 17 SBUV/2, GOME and SCIAMACHY. In the right panel, the time series of NOAA 16 SBUV/2, NOAA 19 SBUV/2, OMI, GOME-2A and GOME-2B for the years 2005 to 2017 are shown. The purpose of these plots is to investigate the consistency, the stability and the
homogeneity of ten completely different time series generated with two different algorithms. It is well shown that, for the





two time periods under consideration, all sensors are in very good agreement, with very similar seasonality amplitudes and biases, further testifying to the homogeneity and stability of the GODFIT v4 products.

## 3 Level-3 Total Ozone Columns

### 3.1 The Level-3 GTO-ECV data record

One of the main aims of the ESA Ozone_cci project is to construct the homogeneous global long-term GOME-type Total Ozone Climate data record, hereafter termed GTO-ECV version3. The individual Level-2 observations (presented and validated above in Section 2) are converted into a Level-3 product and then combined into one single cohesive record spanning the entire 22-years period, from 1995 to 2017. This section summarizes the main characteristics of the merging methodology as well as the latest improvements and extensions implemented within the second phase of the Ozone_cci
project. A detailed description of the predecessor of GTO-ECV v3 has been presented and validated in Loyola et al. (2009) and Coldewey-Egbers et al. (2015).

In short, at first, the individual Level-2 measurements processed with the GODFIT v4 retrieval algorithm are mapped onto a regular global grid of 1°x1° in latitude and longitude to construct daily averages for each sensor. Before combining the individual gridded data, adjustments are made in order to account for possible biases and drifts between the instruments. In
the previous algorithm version, which spanned the 15-years period between March 1996 and June 2011 (Coldewey-Egbers et al., 2015), the GOME TOCs were used as a reference to the other sensors; in this version the OMI measurements serve as a baseline for the inter-sensor calibration. Their long-term stability with respect to ground-based observations data is excellent (see Figure 3 – panels (a) and (c) and Table 1) and the periods of overlap with the other sensors sufficiently long, at least 4 years.

Figure 7 shows the percentage differences between OMI and the other four sensors for 1° zonal monthly mean ozone columns during overlap periods. These zonal means were computed for collocated daily gridded data in order to minimize the impact of differences in the sampling pattern for OMI and the corresponding second sensor. In general, the inter-sensor consistency is very good; mean differences are between -0.2 ± 0.9 % (for GOME-2B, lower right) and 1.0 ± 1.4 % (for SCIAMACHY, upper right). In the inner tropics the bias is slightly negative for all sensors and it increases toward higher
latitudes. The differences between OMI and GOME show slightly larger scatter in the Southern Hemisphere due to significantly reduced spatial coverage of GOME as a consequence of the tape recorder failure in June 2003. The differences between OMI and SCIAMACHY indicate a positive bias for most parts of the Globe, with a maximum in the southern hemisphere around the polar night. For both GOME and SCIAMACHY we apply correction factors using the seasonal mean differences, calculated from the seasonal mean average of all available years, with respect to OMI as a function of latitude.
The differences between OMI and GOME-2A indicate a positive drift of ~ 0.15 % per annum in the middle latitudes of both hemispheres, which we take into account during the adjustment. Likewise for GOME-2B, the correction factors with respect

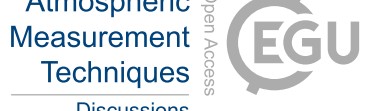

to OMI depend on time (month) and latitude. The adjustment is then applied to the daily gridded data for each individual sensor. Thereby the monthly correction factors are linearly interpolated in time.

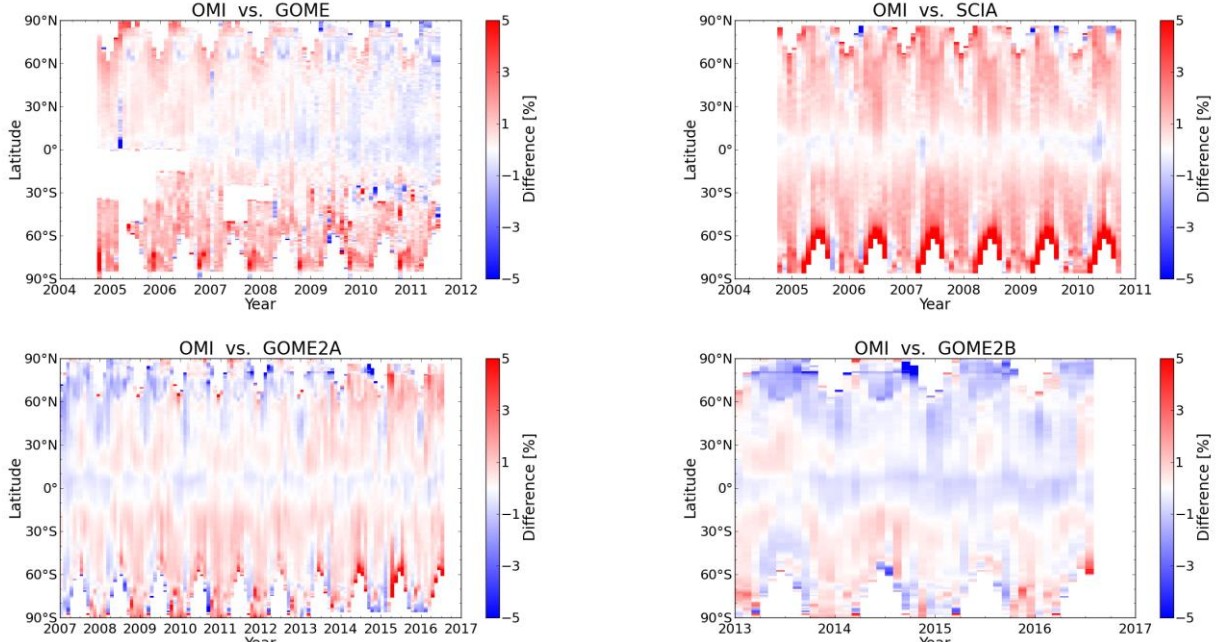

**Figure 7. Percentage differences between OMI and the other four sensors for 1° zonal monthly mean ozone columns during overlap periods. Top left: GOME, top right: SCIAMACHY, bottom left: GOME-2A and bottom right: GOME-2B.**

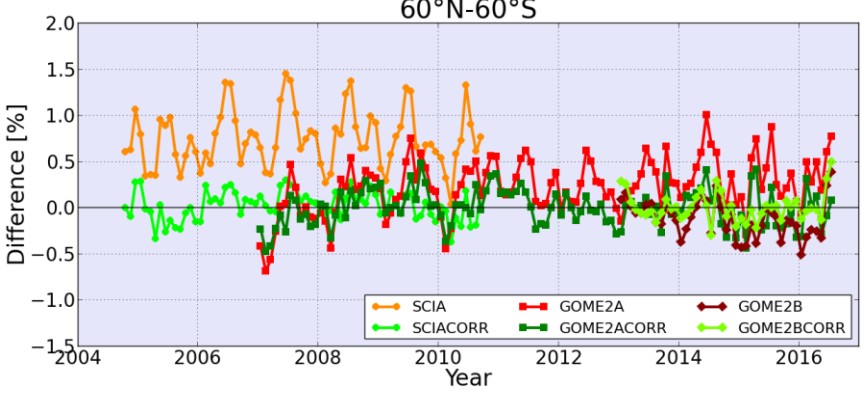

**Figure 8. Percentage differences between SCIAMACHY and OMI (circles), GOME-2A and OMI (squares), and GOME-2B and OMI (diamonds) as a function of time for the periods of overlap. Orange-reddish curves denote the differences without adjustment to OMI, and greenish curves denote the differences after the adjustment to OMI.**





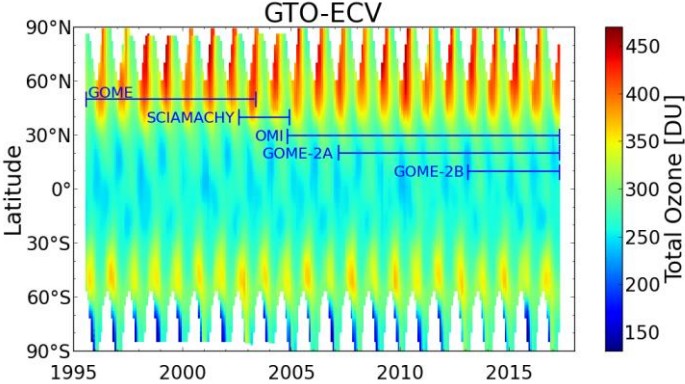

**Figure 9. GTO-ECV total ozone column data record as a function of latitude and time from July 1995 to March 2017. Blue horizontal lines indicate the period for each sensor included in the merged product.**

5    Figure 8 shows the percentage differences between OMI and the other sensors without (orange-reddish curves) and with (greenish curves) the adjustment to OMI for the near global (60° N - 60° S) mean ozone column as a function of time during the periods of overlap. The comparison with GOME is omitted in this plot because we use these data only until June 2003 in the final product. After the application of the correction the mean biases are almost completely reduced, the scatter (standard deviation) decreased by 15 – 40 % and the drift in the differences between GOME-2A and OMI is eliminated.

10    Subsequently, the individual (adjusted) data sets are combined into one single record. In contrast to the previous version (Coldewey-Egbers et al., 2015), where we used only one instrument at any given time, in GTO-ECV v3 we now average all available daily measurements. GOME data are restricted to up and until June 2003. As the ground-based validation of SCIAMACHY Level-2 data indicates some lingering issues with the Level-2 TOCs (see Sec. 2.3) we use SCIAMACHY only until October 2004 in order to fill the data gap between the GOME loss of global coverage and the launch date of OMI.

15    For the calculation of monthly means we apply the same latitudinal constraints as defined in Coldewey-Egbers et al. (2015), see their Table 2, in order to provide representative averages that contain a sufficient number of measurements equally distributed over time. The complete merged GTO-ECV v3 data record with typical ozone characteristics is shown in Figure 9. Highest ozone values occur in northern hemispheric springtime, whereas monthly mean values are below 200 D.U. from September to November southwards of 70° S. Blue horizontal lines indicate the period for each sensor included.

## 3.2    Level-3 validation results and discussion

The validation of the new Level-3 GTO-ECV v3 merged product was performed using as ground truth the Brewer and Dobson spectrophotometer network described in Section 2.2, as was applied in the validation of the previous Level-3 record (Coldewey-Egbers et al., 2015). In order to create the Level-3 TOC field, based on the WOUDC ground-based stations, the





reported TOCs were gridded into the same 1°x1° grid as the GTO-ECV GODFIT data, on a monthly basis, with most grid points being represented by only one reporting station. In detail, direct Sun measurements were considered for the gridding of the ground-based TOCs into Level-3 grid points, even though in some cases this choice severely decreases the number of measurements. As also performed in Coldewey-Egbers et al., 2015, the threshold on the number of measurements available

before the computation of the associated monthly mean was investigated. As a compromise between obtaining the highest global coverage possible and the most representative monthly means, especially at high latitudes, a lower limit of 10 measurements per month and per grid box was enforced so that the temporal representativeness errors are minimized. We note here that restricting the monthly collocated measurements with respect to their mean effective day, which is a measure for the temporal distribution of the daily measurements within a month, did not alter significantly the findings, whereas it

excluded entire zones and months from the comparative process and we opted not to apply such a restriction here.

Figure 10 shows the percentage difference between the satellite and the Brewer (left) and Dobson (right) TOC records as a function of latitude. The five individual satellite TOCs are very consistent with each other for all latitudes and in very close agreement with the ground-based data. The Level-3 comparisons (purple line) show very good agreement with the individual Level-2 lines. In particular, over the NH, all Level-2 comparisons (apart from SCIAMACHY, in green) show a slight

positive deviation of 0 – 2 % to the ground-based data for both ground-based instrument types. In the SH the Level-3 comparisons show a near-perfect agreement with the Level-2 comparisons, apart from the 70° - 80° S belt, where the spread in comparisons reaches the 3.0 % level, which may be attributed to sampling differences between the Level-2 and Level-3 data (see Coldewey-Egbers et al., 2015 for more in-depth discussion of this issue).

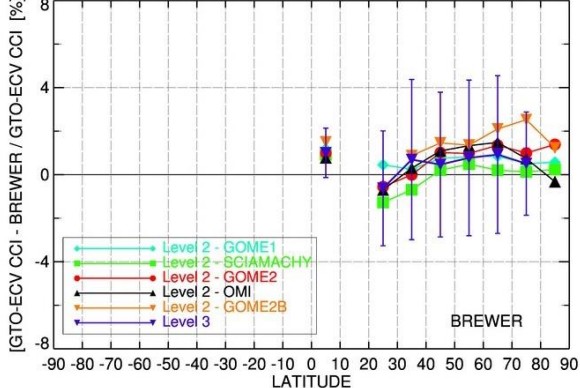 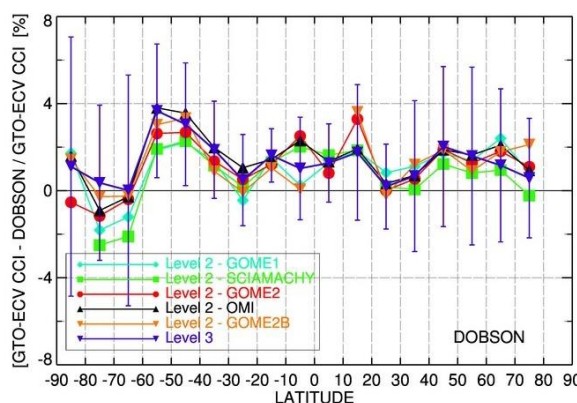

**Figure 10: Latitudinal variability of the percentage difference between satellite observations and ground-based measurements. Left: for the Brewer network and right: for the Dobson network. Light blue line: GOME Level-2 comparison, green line: SCIAMACHY Level-2 comparison, red line: GOME-2A Level-2 comparison, black line: OMI Level-2 comparison, orange line: GOME-2B Level-2 comparison and purple line: Level-3 GTO-ECV v3 comparison. The 1-σ standard deviation of the average is**

**also displayed only for the Level-3 lines.**



In Figure 11 the NH and SH timeseries comparisons of the Level-2 and Level-3 data records with the Dobson and Brewer measurements are shown. The Dobson comparisons for SH (panel a) and NH (panel b) show very good agreement between Level-3 and individual Level-2 lines, within the 1 % difference level for most of the 22-year data record, except for a small

number of outliers. The Brewer comparison in the NH (panel c) shows less amplitude than the Dobson comparisons throughout the full time series, for reasons discussed already in Sections 2.2 and 2.3.

The agreement between the five datasets and the ground-based measurements is excellent, with 0.5 to 1.5 % peak-to-peak amplitude. For the entire time series of the Level-3 data record the mean difference remains mainly positive for all time-series comparisons shown in Figure 11. Concerning the Level-3 comparisons in the NH, the drift per decade of the

differences with respect to ground-based data is negligible, -0.11 ± 0.10 % per decade for Dobson and +0.22 ± 0.08 % per decade for Brewer collocations. Similarly to Level-3, no long-term drift in the differences of the individual Level-2 data sets was found for either Dobson and Brewer comparisons, with OMI showing the smallest drift per decade (in the NH: +0.05 ± 0.12 % for Brewer and -0.39 ± 0.19 % for Dobson, in the SH: -0.15 ± 0.15 % for Dobson measurements). The good quality of the GTO-ECV v3 Level-3 TOC record temporal stability, which well satisfies the requirements for the long term stability

for total ozone measurements of between 1 – 3 % per decade (van der A et al., 2011) and the excellent inter-sensor consistency, make the new Level-3 GTO-ECV v3 dataset suitable and useful for longer term analysis of the ozone layer, such as decadal trend studies (e.g. Coldewey-Egbers et al., 2015), the evaluation of chemistry-climate model projections, and data assimilation applications.

In order to assess and ensure the quality of the new Level-3 GTO-ECV v3 dataset, comparisons are performed against the

solar backscatter ultraviolet (SBUV) merged data product, also shown above in the Level-2 TOC validation section and recently quality assured in Frith et al. (2017). In Figure 12, the time series comparison between GTO-ECV v3 and SBUV merged are presented for the NH and Dobson (panel a), the SH and Dobson (panel b) and the NH and Brewer (panel c) instrument types. The Level-3 GTO-ECV v3 (red line) and SBUV merged (black line) datasets show an excellent agreement of within ± 1.5 %, considering their individual instrumental and algorithm differences, as well as a very similar seasonal

variability with a peak-to-peak amplitude between -1 % and +2 % in Dobson and -0.5 % and +1 % in Brewer cases over the entire time period. Furthermore, the two datasets show almost the same negligible drift per decade in the NH for both ground-based instrument networks, whereas in the SH for Dobson collocations the drift per decade is +0.23 ± 0.09 % and -0.09 ± 0.07 % for the Level-3 GTO-ECV v3 and the SBUV merged TOCs, respectively.



(a)

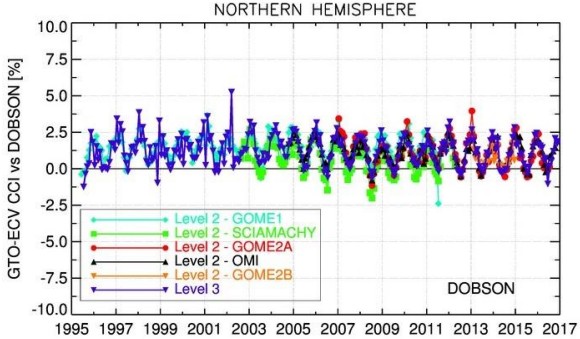

(b)

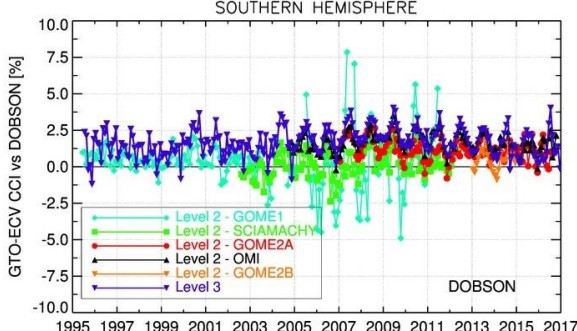

(c)

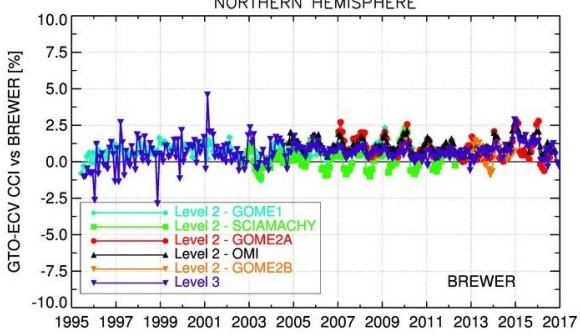

**Figure 11: Time series of the percentage difference between satellite observations and ground-based measurements for the Dobson network in the NH (panel a) and in the SH (panel b) and for the Brewer network, NH only (panel c). Light blue line: GOME Level-2 comparison, green line: SCIAMACHY Level-2 comparison, red line: GOME-2A Level-2 comparison, black line: OMI Level-2 comparison, orange line: GOME-2B Level-2 comparison and purple line: Level-3 GTO-ECV v3 comparison.**



(a)

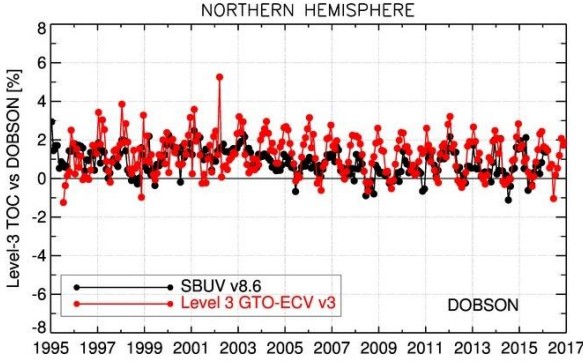

(b)

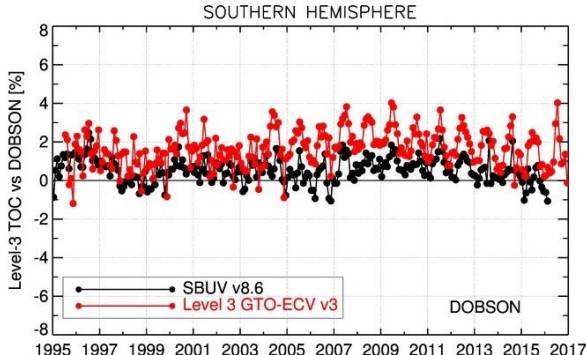

(c)

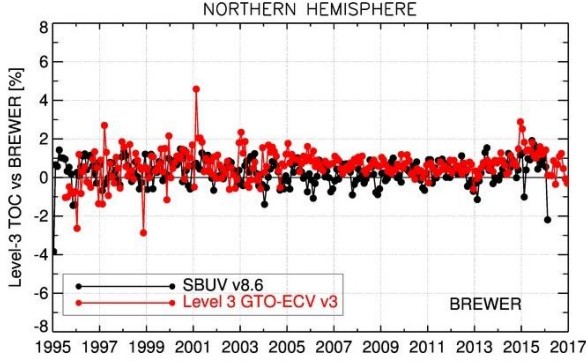

Figure 12: Same as in Figure 11. Black line: SBUV merged comparison and red line: Level-3 GTO-ECV v3 comparison.



## 4    Summary and conclusions

In this work, the Essential Climate Variable (ECV) Climate Research Data Package Total Ozone Column (CRDP TOC), refined and updated via the European Space Agency's *Climate Change Initiative* Phase-II, is presented and validated against independent ground-based TOC observations. Level-2 TOCs, produced by the GODFIT v4 algorithm as applied to the GOME/ERS-2, OMI/Aura, SCIAMACHY/Envisat and GOME-2/MetopA and /MetopB observations, form the basis for a 22-year long consistent, smooth and homogeneous CRDP. In addition the individual sensor products have been combined and merged into one single cohesive Level-3 data record, GTO-ECV v3. Detailed quality control and assurance against specific requirements from the international climate-chemistry modelling community showed that the product more than meets the official User Requirements, i.e. that the stability of the TOC measurements has to be between 1 and 3 % per decade, that the radiative forcing introduced by the evolution of the ozone layer has to be less than 2 % and that the short-term variability has to be less than 3 %. In detail:

- The individual Level-2 data sets show excellent inter-sensor consistency with mean differences within 1.0 % at moderate latitudes (+/-50°), whereas the Level-3 data sets show mean differences with respect to the OMI reference data record that span between -0.2 ± 0.9 % (for GOME-2B) and 1.0 ± 1.4 % ( for SCIAMACHY).
- For the Level-2 validation against ground-based measurements: the mean bias between GODFIT v4 satellite and Brewer, Dobson and SAOZ – reported TOCs is well within 1.5 ± 1.0 % for all sensors, the drift per decade spans between 0 % to 1.4 ± 1.0 % depending on the sensor and the peak-to-peak seasonality ranges between ~1 % for GOME and OMI, to ~2 % for SCIAMACHY.
- For the Level-3 validation against ground-based measurements: an excellent agreement with 0.5 to 1.5 % peak-to-peak amplitude for the monthly mean time series is found as well as negligible drift in the Northern Hemisphere differences at -0.11 ± 0.10 % per decade for Dobson and +0.22 ± 0.08 % per decade for Brewer collocations.

We hence conclude that the exceptional quality of the GTO-ECV v3 Level-3 TOC record temporal stability satisfies well the requirements of 1 – 3 % per decade. The excellent inter-sensor consistency renders both the Level-2 GODFIT v4, as well as the Level-3 GTO-ECV v3 datasets, suitable and useful for longer term analysis of the ozone layer, such as decadal trend studies, the evaluation of model simulations, and data assimilation applications.

The Ozone_cci CRDP includes data products for total ozone columns, ozone profiles from nadir sensors and stratospheric ozone profiles from limb and occultation sensors. All data sets are reported in netCDF-CF format following CCI and GCOS standards, and are freely available on the Ozone_cci web site (http://www.esa-ozone-cci.org/?q=node/160).

## Acknowledgements

The authors are grateful to ESA's Climate Change Initiative – Ozone project, Phase II, for providing the support and funding necessary for this work. The ground-based data used in this publication were obtained as part of WMO's Global Atmosphere





Watch (GAW) and the Network for the Detection of Atmospheric Composition Change (NDACC). They are publicly available via the World Ozone and UV Data Centre (WOUDC) and the NDACC Data Host Facility (see http://woudc.org and http://ndacc.org, respectively). We would like to acknowledge and warmly thank all the investigators that provide data to these repositories on a timely basis, as well as the handlers of these databases for their upkeep and quality guaranteed efforts. We would also like to acknowledge the SBUV instrument team members for their work producing SBUV version 8.6 TOC data record.

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
