# Peer review of "Quality assessment of the Ozone\_cci Climate Research Data Package (release 2017): 1. Ground-based validation of total ozone column data products"

_Atmospheric Measurement Techniques, 2017_

## Referee Comment (RC1) · Anonymous Referee #1 · 21 Nov 2017

General Comments:

Data records of satellite borne instruments are only temporary in contrast to most of the ground based total ozone column (TOC) records. Thus the development of a method to compare the available satellite records and to merge them to create a long term, homogeneous TOC data set, is a very valuable contribution to the monitoring of the ozone layer. This publication gives a very good description of the validation of such merged data records with ground based records of Dobson, Brewer and SAOZ instruments.

Specific Comments:

1. It should be mentioned that the used Dobson and Brewer TOC data records are still based on the "old" Bass and Paur ozone cross sections, whereas it seems that the satellite data are produced using the new ozone cross sections (Bremen, IUP?), good place for this explanation would be page 7 after line 25.

2. Dependence on effective temperature of the Dobsons (p 5- 6): Basher 1982 is not an appropriate reference, as it was written, when the ozone cross-sections after Vigroux had been valid. Current data sets are processed using Bass and Paur. Better and up to date references for this issue are: Koukouli et al., 2016 (cited later in the text, page 7) Scarnato et al., 2009: Temperature and slant path effects in Dobson and Brewer total ozone measurements, Journal of Geophysical Research: Atmospheres, Vol. 114, Issue D24 Kerr, J. B., I. A. Asbridge, and W. F. J. Evans, Intercomparison of total ozone measured by the Brewer and Dobson spectrophotometers at Toronto, J. Geophys. Res., 93, 11,129– 11,140, 1988. Kerr, 2002, New methodology for deriving total ozone and other atmospheric variables from Brewer spectrophotometer direct sun spectra, JOURNAL OF GEOPHYSICAL RESEARCH, VOL. 107, NO. D23

3. The use of SAOZ might be seen a little bit problematically with its accuracy of 6% (page 6)

4. On page 12 a correction for the Izana record due to the altitude is mentioned. Such a correction should make sense for other mountain stations too, especially when they are more or less isolated compared with the 150km footprint of the satellite data. A first guess of correction would be +0.1% per 100m difference of station altitude and environmental altitude. There are some mountain stations with significant differences (e.g. Arosa, Hohenpeissenberg, Mauna Loa). This information can be included in the tables S1 – S3.

5. Addition information in these tables about the lengths of the records would be informative, as not all stations have measured from 1995 to 2017.

6. The explanation on page 9, why the SZA-dependence for the Dobsons are not drawn is misleading. As reason a high correlation between Dobsons' large stratospheric effective temperature dependence and the SZA is mentioned. This correlation is physically not correct. The SZA of daily means of TOC is larger during winter season, when the sun is not very high. In addition in winter the Teff is lower than the used -46 degree Celsius. Thus it is an indirect correlation, which is e.g. not valid during summer season, when Teff is "normal" and Dobson TOCs drop at very high SZA (mue > than 3.5 depending on turbidity) -values because of straylight effects but not because of temperature dependence. In any case it is justified not to use Dobson data at SZA larger than 75 degrees, even if they were available.

7. In figures 4, 5 and 10 Brewer observations are drawn above SZA of 75 degrees. The slant path mue of these measurements are larger than 3.5. Observations with larger mue-values are not accurate enough, especially when using single Brewers. Double Brewers might be able to measure up mue = 4, before the TOC drops (reason see Dobson explanation of straylight effects in the bullet point before).

8. Concerning the seasonality of SAOZ-difference mentioned on page 9 and seen in figure 3: Is there an explanation for this pattern?

Technical corrections:

1. In references Serdyuchenko on page 26 "‐ Part 2" is written instead of "- Part 2". 2. Kerr et al. 1988 is cited on page 5, line 18, but cannot be found in the references.

---

## Referee Comment (RC2) · Anonymous Referee #2 · 18 Dec 2017

The manuscript presents the validation study for the merged satellite total ozone record obtained from the GOME-type satellite ozone sensors. In this work the new dataset is validated against the ground-based network of Dobson, Brewer an SAOZ instruments. Presented study fits well to the scope of the problems covered in the AMT. The manuscript is well written and organized.

Major comments:

-Most of the results presented in the study are shown for the hemispheric monthly

means. However, the description of the methodology to compute these hemispheric means is not provided. These means could be computed in a number of ways (e.g. with/without weights), thus it would be important to provide a brief description in the text or in the appendix/supplement. Scientific results should be reproducible, and the clear description of the methods is an important component to ensure the robustness of the results.

-In the Tables in the Supplement, please, indicate the time periods for which the data from individual ground-based stations were used in this study.

-Page 9, lines 15-18: I don't quite understand the reason for not showing a plot with the SZA dependence for Dobson comparisons. Authors stated that Dobson observations depend on the stratospheric temperature, which should produce an artificial dependence on SZA. At the same time, when authors discuss results for comparisons with Brewer and SAOZ, they claim that the observed dependence on SZA for satellite measurements doesn't matter because all satellites show consistent patterns relative to ground-based stations. If so, why don't to show results with Dobson if the goal is to check consistency among satellite records.

Page 12, line 15: authors mentioned that the results of the comparison with measurements at Izana station were adjusted to account for the station's elevation. Have these adjustments been applied to any other station? This should be clearly described in the text.

- Page 12-13, Table 1: I found that the quantities shown in Table 1 are not well described. I think this part needs some major revision, including terms that are used in the table. For example, I would recommend using a term "mean bias" instead of "monthly mean bias" and mentioning in the text that the mean bias was computed from monthly mean differences, because to me "monthly mean bias" would mean the bias in a specific month, while, if I understood this correctly, the biases shown in Table 1 were computed over the entire data record. I don't quite understand what is "monthly mean

variability" in Table 1. The 1–sigma standard deviations for biases are shown along with the biases. In the text this quantity is explained as "the variability of the monthly mean standard deviation values". Is that the variability of the standard deviations of differences in individual months? I would also recommend replacing "Seasonality" with "seasonal bias". The two last lines in the table are very confusing (Latitude and SZA). At first, I thought they show the mean differences in latitude and sza between satellite and ground based observations. But according to the text these are the mean differences in ozone between satellite and ground-based observations, calculated by averaging all points shown in Figure 4 and Figure 5. I understand that if you try to bin differences in smaller bins (like you did with SZA) you can uncover some dependence on SZA. Then it would make sense to name these quantities as "Latitudinal biases" and "SZA biases". But why do you expect results to be different from the mean biases if you average over the entire latitude range? Please, explain.

-Page 15 lines 25-31 and Page 16 lines 1-2: In this part of the manuscript authors describe the correction factors against OMI that were used to correct individual satellite time series. It is stated that "we apply correction factors using the seasonal mean differences...". Then it mentioned that the drift in GOME 2A have been accounted for. Is it a static correction that depends on lat/lon and month of the year only? Or have you implemented time-dependent corrections? Please, explain that in the text.

-Page 17, lines 10-20: you need to explain what was done in the merging process when data from two or more instruments are available. Did you simply average all available data? Did you use some weights?

-Looking at the results showing in Figure 11, it seems to me that the merged dataset almost fully overlaps with OMI. This is expected since all individual datasets have been corrected against OMI, and OMI has a very dense spatial and temporal coverage. My question here: what is the value of using GOME 2A or GOME 2B in the merged product? Please, provide an explanation in the text.

[Figure]

-Figure 7: I would suggest to keep the range for the time scale (X-axis) the same for all 4 panels;

-Figure 9: it's very hard to see blue letters on the green background. I would suggest to move satellite timelines either to the top or bottom of the figure.

-Figure 10, left panel: why there is no point for Level 3 product in 80-90N latitude bin even though data for all individual instruments are shown;

-I am puzzled why the Level 3 value in 70-80S latitude bin is higher then for any given individual instrument.

Minor comments:

- All abbreviations should be spelled out when used for the first time in the text. For instance, there are many abbreviations in the Abstract and Introduction that are not explained: P. 1, line 15: "GOME-type" – please, spell out "GOME"; P. 1, line 18: "GOD-FIT", "ERS", "OMI", "SCIAMACHY" –please, spell them out; P. 2, line 33: please, spell out "SAOZ"; P. 3, line 7: please, spell out "BIRA-IASB" and "DLR"; P. 3, line 10: please, spell out "LIDORT"

- P.3 lines 25-28: It is not quite clear from the context which quantity has "been estimated to rise up to +/-2%": systematic uncertainty in the ozone cross sections or ozone itself? Please, consider re-wording this statement.

- P. 14, line 23: I guess it should be "NOAA 18" (not NOAA 16) to match with the labels on the right panel of Figure 6.

-There are several places in the manuscript where authors use words "excellent", "exceptional" etc. I would recommend to avoid these statements in the scientific publication and rather provide quantitative results like "the stability within +/-1%" or "biases less than 2%".

---

## Author Comment (AC1) · 22 Jan 2018

##

**General Comments:**

Data records of satellite borne instruments are only temporary in contrast to most of the ground based total ozone column (TOC) records. Thus the development of a method to compare the available satellite records and to merge them to create a long term, homogeneous TOC data set, is a very valuable contribution to the monitoring of the ozone layer. This publication gives a very good description of the validation of such merged data records with ground based records of Dobson, Brewer and SAOZ instruments.

**Specific Comments:**

1. **Comment:**
   It should be mentioned that the used Dobson and Brewer TOC data records are still based on the "old" Bass and Paur ozone cross sections, whereas it seems that the satellite data are produced using the new ozone cross sections (Bremen, IUP?), good place for this explanation would be page 7 after line 25.
   **REPLY:**
   The explanation is added in section 2.3, as suggested. Thank you.

2. **Comment:**
   Dependence on effective temperature of the Dobsons (p 5- 6): Basher 1982 is not an appropriate reference, as it was written, when the ozone cross-sections after Vigroux had been valid. Current data sets are processed using Bass and Paur. Better and up to date references for this issue are: Koukouli et al., 2016 (cited later in the text, page 7) Scarnato et al., 2009: Temperature and slant path effects in Dobson and Brewer total ozone measurements, Journal of Geophysical Research: Atmospheres, Vol. 114, Issue D24 Kerr, J. B., I. A. Asbridge, and W. F. J. Evans, Intercomparison of total ozone measured by the Brewer and Dobson spectrophotometers at Toronto, J. Geophys. Res., 93, 11,129– 11,140, 1988. Kerr, 2002, New methodology for deriving total ozone and other atmospheric variables from Brewer spectrophotometer direct sun spectra, JOURNAL OF GEOPHYSICAL RESEARCH, VOL. 107, NO. D23
   **REPLY:**
   The references are added. Thank you for the suggestions.

3. **Comment:**

The use of SAOZ might be seen a little bit problematically with its accuracy of 6% (page 6)

**REPLY:**

While this overall accuracy is poor in comparison to that of the direct-sun instruments, the added value provided by the SAOZ instruments is their ability to produce reference measurements at those locations and times-of-year where and when the satellite measurements occur under low-sun conditions and no reliable direct-sun measurements can be made. As such, they allow the validation of an otherwise inaccessible satellite measurement regime. This point was already made in the paper.

Moreover, it must also be noted that a significant fraction of this 6% total accuracy is made up of the (systematic) uncertainty in the $O_3$ cross sections (3%) and by the impact of clouds (3.3 %, Hendrick et al., 2011), both of which are of minor importance in differential analyses of cloud-free data. This note was added in the paper, in Section 2.2.

4. **Comment:**

On page 12 a correction for the Izana record due to the altitude is mentioned. Such a correction should make sense for other mountain stations too, especially when they are more or less isolated compared with the 150km footprint of the satellite data. A first guess of correction would be +0.1% per 100m difference of station altitude and environmental altitude. There are some mountain stations with significant differences (e.g. Arosa, Hohenpeissenberg, Mauna Loa). This information can be included in the tables S1 – S3.

**REPLY:**

The mentioned correction for the SAOZ measurements is an ERA-Interim-based estimate of the column below the instrument altitude in the immediate vicinity of the island and/or mountain top (see Verhoelst et al, 2015 for further details). For the SAOZ/ZSL-DOAS network, Izana and Jungfraujoch are the only stations for which a significant missing column was derived with this methodology (about 2.8% and 3.2% respectively, with some seasonal variation), due to their isolated mountain-top locations.

As for the ground based measurements performed by Dobson and Brewer spectrophotometers that are used in this work, since they are downloaded from the WOUDC database we are not able to correct them for the altitude issue, as suggested. Nevertheless, we can use the information to identify any discrepancies seen in our figures.

Furthermore, as seen in Koukouli et al. (2016), when a high altitude station like Hohenpeissenberg (where the gradient is not very steep and the instrument is exceptionally maintained and calibrated) is used, the satellite-to-ground comparison is excellent (Brewer bias ~0.3% and Dobson bias ~1%, see figure below). For the Mauna Loa station (10° - 20° N), on the other hand, where the gradient is much steeper, the satellite-to-ground comparison is about 2-4%. However, when considering zonal means of the differences, where all available stations in each belt are included in the calculations, the effect of the station altitude becomes less evident, which is the case for the 10-20° N belt in Figure 5 – panel (a) where Mauna Loa and Bangkok are co-calculated.

Thank you for the suggestion about this issue, we will take it seriously under consideration and use it as basis for a future study.

Some more information on the SAOZ measurements' correction is added in the manuscript, in section 2.3. We have also added the altitude information for each station in the Tables S1 – S3.

[Figure]

Koukouli et al. (2016) – Figure 1.

**5. Comment:**

Addition information in these tables about the lengths of the records would be informative, as not all stations have measured from 1995 to 2017.

**REPLY:**

We have added the time period for each ground based station in the Tables S1 – S3.

**6. Comment:**

The explanation on page 9, why the SZA-dependence for the Dobsons are not drawn is misleading. As reason a high correlation between Dobsons' large stratospheric effective temperature dependence and the SZA is mentioned. This correlation is physically not correct. The SZA of daily means of TOC is larger during winter season, when the sun is not very high. In addition in winter the Teff is lower than the used -46 degree Celsius. Thus it is an indirect correlation, which is e.g. not valid during summer season, when Teff is "normal" and Dobson TOCs drop at very high SZA (mue> than 3.5 depending on turbidity) -values because of straylight effects but not because of temperature dependence. In any case it is justified not to use Dobson data at SZA larger than 75 degrees, even if they were available.

**REPLY:**

Thank you for the suggestion. We agree with the comment and we have added two plots (SH and NH) in Figure 4 (and the respective comments) showing the dependence of the satellite-to-Dobson comparison on SZA. As for the cut-off at 75°, we did not apply it because the SZAs used for the binning and the plots are the satellite SZAs, since we use daily means of the ground based measurements. We have also added a sentence in section 2.3 making this clear.

**7. Comment:**

In figures 4, 5 and 10 Brewer observations are drawn above SZA of 75 degrees. The slant path mue of these measurements are larger than 3.5. Observations with larger mue-values

are not accurate enough, especially when using single Brewers. Double Brewers might be able to measure up mue = 4, before the TOC drops (reason see Dobson explanation of straylight effects in the bullet point before).

**REPLY:**

Thank you for the suggestion. As explained in the previous comment, please note that the SZAs used for the binning and the plots are the satellite SZAs, since we use daily means of the ground based measurements.

8. **Comment:**

Concerning the seasonality of SAOZ-difference mentioned on page 9 and seen in figure 3: Is there an explanation for this pattern?

**REPLY:**

It should be noted that seasonality seen on Figure 3 are observed at all latitudes and all instruments but,

   i. the amplitude is larger in NH compared to SH on both SAOZ and Dobson
   ii. the amplitude is larger with SAOZ compared to Dobson and Brewer
   iii. the amplitude varies with the satellites, the largest being with GOME and SCIAMACHY in Northern hemisphere.
   iv. the strongest minima are observed in the winter particularly on the difference between SAOZ and GOME.

The seasonality can be attributed to:

   i) the cross sections dependencies on the effective temperature of the stratosphere impacting all measurements in the UV but not SAOZ analyzing in the visible.
   ii) the number of stations used in the statistics in winter, limited in latitudes for Dobson and Brewer but being possible at higher latitude for SAOZ.

The SAOZ seasonality observed on panels (d) and (e) of Figure 3 comes from the latitude of the selected stations, which, in the case of SAOZ, allows to perform comparisons in winter at high latitude when the effect of the temperature on UV cross section is the largest.

We have modified the respective paragraph commenting on panels (d) and (e) of Figure 3, in Section 2.3, to give a more clear explanation of the seasonality effect.

**Technical corrections:**

1. In references Serdyuchenko on page 26 "‐ Part 2" is written instead of "- Part 2".
2. Kerr et al. 1988 is cited on page 5, line 18, but cannot be found in the references.

**REPLY:**

The references are corrected/added. Thank you!

---

## Author Comment (AC2) · 22 Jan 2018

The manuscript presents the validation study for the merged satellite total ozone record obtained from the GOME-type satellite ozone sensors. In this work the new dataset is validated against the ground-based network of Dobson, Brewer an SAOZ instruments. Presented study fits well to the scope of the problems covered in the AMT. The manuscript is well written and organized.

**Major comments:**

1. **Comment:**
Most of the results presented in the study are shown for the hemispheric monthly means. However, the description of the methodology to compute these hemispheric means is not provided. These means could be computed in a number of ways (e.g. with/without weights), thus it would be important to provide a brief description in the text or in the appendix/supplement. Scientific results should be reproducible, and the clear description of the methods is an important component to ensure the robustness of the results.
**REPLY:**
Thank you for this remark. We agree and we have added a few sentences explaining the methodology of our calculations in Section 2.3, where Figure 3 is commented.

2. **Comment:**
In the Tables in the Supplement, please, indicate the time periods for which the data from individual ground-based stations were used in this study.
**REPLY:**
We have added the time period for each ground based station in the Tables S1 – S3.

3. **Comment:**
Page 9, lines 15-18: I don't quite understand the reason for not showing a plot with the SZA dependence for Dobson comparisons. Authors stated that Dobson observations depend on the stratospheric temperature, which should produce an artificial dependence on SZA. At the same time, when authors discuss results for comparisons with Brewer and SAOZ, they claim that the observed dependence on SZA for satellite measurements doesn't matter because all satellites show consistent patterns relative to ground-based stations. If so, why don't to show results with Dobson if the goal is to check consistency among satellite records.

**REPLY:**
Thank you for the suggestion. We agree with the comment and we have added two plots (SH and NH) in Figure 4, showing the dependence of the satellite to Dobson comparison on SZA, and the respective comments in section 2.3.

4. **Comment:**
Page 12, line 15: authors mentioned that the results of the comparison with measurements at Izana station were adjusted to account for the station's elevation. Have these adjustments been applied to any other station? This should be clearly described in the text.
**REPLY:**
The mentioned correction is an ERA-Interim-based estimate of the column below the instrument altitude in the immediate vicinity of the island and/or mountain top (see Verhoelst et al, 2015, for further details). For the SAOZ/ZSL-DOAS network, Izana and Jungfraujoch are the only stations for which a significant missing column was derived with this methodology (about 2.8% and 3.2% respectively, with some seasonal variation), due to their isolated mountain-top locations. Some more information is added in the manuscript, section 2.3.

5. **Comment:**
Page 12-13, Table 1: I found that the quantities shown in Table 1 are not well described. I think this part needs some major revision, including terms that are used in the table. For example, I would recommend using a term "mean bias" instead of "monthly mean bias" and mentioning in the text that the mean bias was computed from monthly mean differences, because to me "monthly mean bias" would mean the bias in a specific month, while, if I understood this correctly, the biases shown in Table 1 were computed over the entire data record. I don't quite understand what is "monthly mean variability" in Table 1. The 1–sigma standard deviations for biases are shown along with the biases. In the text this quantity is explained as "the variability of the monthly mean standard deviation values". Is that the variability of the standard deviations of differences in individual months? I would also recommend replacing "Seasonality" with "seasonal bias". The two last lines in the table are very confusing (Latitude and SZA). At first, I thought they show the mean differences in latitude and sza between satellite and ground based observations. But according to the text these are the mean differences in ozone between satellite and ground-based observations, calculated by averaging all points shown in Figure 4 and Figure 5. I understand that if you try to bin differences in smaller bins (like you did with SZA) you can uncover some dependence on SZA. Then it would make sense to name these quantities as "Latitudinal biases" and "SZA biases". But why do you expect results to be different from the mean biases if you average over the entire latitude range? Please, explain.
**REPLY:**
Thank you for the comments and the suggestions. Please find below our replies:
- The "Monthly mean bias" term was corrected in the text as suggested and a phrase explaining how it was calculated is added.
- The "monthly mean variability" term is indeed the variability of the monthly mean standard deviation values and the text is modified so as to be clearer.

- "Seasonality": We consider the "bias" as a term to be the deviation of comparisons from the 0% line, which would mean 0% bias. In this statistic quantity we calculate the peak-to peak range in values, so we think changing the term would imply a different statistical quantity.
- Latitude and SZA statistics: we agree that we do not expect to see any major differences to the mean differences of the monthly means, but these statistical quantities have to be provided as requirements by the Users of the ESA CCI project (they will be uploaded soon at: http://www.esa-ozone-cci.org/?q=documents# ). We have added the reference and changed the text accordingly. We also named the quantities "Latitudinal mean bias" and "SZA mean bias".

6. **Comment:**
Page 15 lines 25-31 and Page 16 lines 1-2: In this part of the manuscript authors describe the correction factors against OMI that were used to correct individual satellite time series. It is stated that "we apply correction factors using the seasonal mean differences…". Then it mentioned that the drift in GOME 2A have been accounted for. Is it a static correction that depends on lat/lon and month of the year only? Or have you implemented time-dependent corrections? Please, explain that in the text.
**REPLY:**
For GOME and SCIAMACHY it is a static correction that depends on latitude and month of the year. For GOME-2A and GOME-2B it is a fully time- (and latitude-) dependent correction. We have improved the corresponding explanation in the text, Section 3.1.

7. **Comment:**
Page 17, lines 10-20: you need to explain what was done in the merging process when data from two or more instruments are available. Did you simply average all available data? Did you use some weights?
**REPLY:**
The merging is done on a daily basis. When data from two or more instruments are available, we average all and use as weights the number of measurements per day and grid box for the corresponding sensor. We added the explanation in Section 3.1.

8. **Comment:**
Looking at the results showing in Figure 11, it seems to me that the merged dataset almost fully overlaps with OMI. This is expected since all individual datasets have been corrected against OMI, and OMI has a very dense spatial and temporal coverage. My question here: what is the value of using GOME 2A or GOME 2B in the merged product? Please, provide an explanation in the text.
**REPLY:**
We agree with the reviewer that it is not unexpected that the results of the level-3 validation are dominated by the very dense OMI measurements. As already seen in the ground-based validation of the previous version of the merged product (Coldewey-Egbers et al., 2015) the quality of monthly mean level-3 data strongly depends on the spatio-temporal coverage of the input (level-2) measurements. Therefore, we think that it is more beneficial to include all

available measurements in order to further improve coverage and, thus, statistics and representativeness of the monthly mean values. We added the explanation in Section 3.1.

9. **Comment:**
Figure 7: I would suggest to keep the range for the time scale (X-axis) the same for all 4 panels;
**REPLY:**
We changed the range for the time scale as suggested and we added a gray background to better distinguish between missing data and values close to zero.

10. **Comment:**
Figure 9: it's very hard to see blue letters on the green background. I would suggest to move satellite timelines either to the top or bottom of the figure.
**REPLY:**
We changed the color of the letters to 'black' and moved the timelines to the middle of the plot in order to improve visibility.

11. **Comment:**
Figure 10, left panel: why there is no point for Level 3 product in 80-90N latitude bin even though data for all individual instruments are shown;
**REPLY:**
Thank you for noticing this issue. The reason is that there is one ground-based station nearly at the limit between the 70°- 80° bin and the 80°- 90° bin, namely Eureka, at 79°.89 North. As a result, when allowing spatial collocations to the satellite central pixel within a certain radius, some of the collocations were allocated to the 80°- 90° bin.
We have updated our validation chain to take this into account and changed Figure 10 accordingly.
When analyzing the ground-based level-2 into gridded level-3 this issue did not come up, so there was from the start no data in the 80° -90° bin.

12. **Comment:**
I am puzzled why the Level 3 value in 70-80S latitude bin is higher than for any given individual instrument.
**REPLY:**
The time series shown in this Figure are not common collocations, as you can well imagine, since the different Level-2 instruments have a different time span, which again differs from the level-3 data. Hence, the variability shown by all six time series in the -70° to -80° S bin may be due to a number of factors, including ground-based data availability, as well as dynamic issues that may affect different years in a different manner, such as the polar vortex for example.

**Minor comments:**

1. All abbreviations should be spelled out when used for the first time in the text. For instance, there are many abbreviations in the Abstract and Introduction that are not explained: P. 1, line 15: "GOME-type" – please, spell out "GOME"; P. 1, line 18: "GODFIT", "ERS", "OMI",

"SCIAMACHY" –please, spell them out; P. 2, line 33: please, spell out "SAOZ"; P. 3, line 7: please, spell out "BIRA-IASB" and "DLR"; P. 3, line 10: please, spell out "LIDORT"

**REPLY:**

We have spelled all abbreviations in the main text (mostly in the introduction), as suggested. The satellite names in the Abstract are left as they were because we think that it would make it too extensive, but they are also spelled in the introduction.

2. P.3 lines 25-28: It is not quite clear from the context which quantity has "been estimated to rise up to +/-2%": systematic uncertainty in the ozone cross sections or ozone itself? Please, consider re-wording this statement.

   **REPLY:**

   We agree that it was not quite clear in the text that the quantity that is biased by +/-2 % due to the use of different cross sections, is ozone. The manuscript was rephrased. Thank you for the suggestion.

3. P. 14, line 23: I guess it should be "NOAA 18" (not NOAA 16) to match with the labels on the right panel of Figure 6.

   **REPLY:**

   Yes, thank you, it should be NOAA 18. We have made the correction.

4. There are several places in the manuscript where authors use words "excellent", "exceptional" etc. I would recommend to avoid these statements in the scientific publication and rather provide quantitative results like "the stability within +/-1%" or "biases less than 2%".

   **REPLY:**

   Thank you, we have scanned through the text and made all the necessary alterations.